# Effects of Selenium Yeast on Egg Quality, Plasma Antioxidants, Selenium Deposition and Eggshell Formation in Aged Laying Hens

**DOI:** 10.3390/ani13050902

**Published:** 2023-03-01

**Authors:** Zhexi Liu, Yutao Cao, Yue Ai, Gang Lin, Xiaonan Yin, Linli Wang, Mengyao Wang, Bingkun Zhang, Keliang Wu, Yuming Guo, Hongbing Han

**Affiliations:** 1Beijing Key Laboratory of Animal Genetic Improvement, College of Animal Science and Technology, China Agricultural University, Beijing 100193, China; 2National Engineering Laboratory for Animal Breeding, College of Animal Science and Technology, China Agricultural University, Beijing 100193, China; 3Key Laboratory of Animal Genetics, Breeding and Reproduction of the Ministry of Agriculture and Rural Affairs, College of Animal Science and Technology, China Agricultural University, Beijing 100193, China; 4Institute of Quality Standards and Testing Technology for Agricultural Products, Chinese Academy of Agricultural Sciences, Beijing 100081, China; 5Beijing Alltech Biological Products (China) Co., Ltd., Beijing 100600, China; 6State Key Laboratory of Animal Nutrition, College of Animal Science and Technology, China Agricultural University, Beijing 100193, China

**Keywords:** selenium yeast, eggshell quality, transcriptomic analysis, aged laying hens

## Abstract

**Simple Summary:**

The depression of egg quality during the late stage of the laying cycle is one of the knotty issues in the poultry industry, which cause a great economic loss. Our study revealed that 0.45 mg/kg selenium yeast supplementation can alleviate the decline of egg quality in aged laying hens. In addition, organs Se levels and plasma antioxidant capacity were significantly higher with Se supplementation. Transcriptomic analysis identified some key candidate genes and potential molecular processes involved in selenium yeast’s effects on eggshell formation. We inferred that 0.45 mg/kg selenium yeast has potential functions to alleviate the decrease in eggshell quality in aged laying hens to extend the laying cycle. These findings provide several new fundamental insights into improving egg quality in aged laying hens and extending the laying cycle.

**Abstract:**

Internal egg and eggshell quality are often deteriorated in aging laying hens, which causes huge economic losses in the poultry industry. Selenium yeast (SY), as an organic food additive, is utilized to enhance laying performance and egg quality. To extend the egg production cycle, effects of selenium yeast supplementation on egg quality, plasma antioxidants and selenium deposition in aged laying hens were evaluated. In this study, five hundred and twenty-five 76-week-old Jing Hong laying hens were fed a selenium-deficient (SD) diet for 6 weeks. After Se depletion, the hens were randomly divided into seven treatments, which included an SD diet, and dietary supplementation of SY and sodium selenite (SS) at 0.15, 0.30, and 0.45 mg/kg to investigate the effect on egg quality, plasma antioxidant capacity, and selenium content in reproductive organs. After 12 weeks of feeding, dietary SY supplementation resulted in higher eggshell strength (SY0.45) (*p* < 0.05) and lower shell translucence. Moreover, organs Se levels and plasma antioxidant capacity (T-AOC, T-SOD, and GSH-Px activity) were significantly higher with Se supplementation (*p* < 0.05). Transcriptomic analysis identified some key candidate genes including cell migration inducing hyaluronidase 1 (*CEMIP*), ovalbumin (*OVAL*), solute carrier family 6 member 17 (*SLC6A17*), proopiomelanocortin (*POMC*), and proenkephalin (*PENK*), and potential molecular processes (eggshell mineralization, ion transport, and eggshell formation) involved in selenium yeast’s effects on eggshell formation. In conclusion, SY has beneficial functions for eggshell and we recommend the supplementation of 0.45 mg/kg SY to alleviate the decrease in eggshell quality in aged laying hens.

## 1. Introduction

Maintaining egg quality and production performance in aged laying hens for longer laying cycles has become an increasing concern. In general, the peak production of commercial laying hens lasts until around 60 weeks of age, and the laying rate and egg quality begin to decline thereafter [1]. Aging of hens leads to deterioration in internal and shell quality of eggs, which may lead to breakage during collection and transport [2]. Thus, overcoming the negative effects of age on egg quality is important in the late phase of the laying cycle. Some previous studies suggested that the decline in egg quality is associated with nutrient status and oxidative stress in aged laying hens [2]. Selenium (Se) is an essential micronutrient for humans and animals which plays a vital role in antioxidant defense, redox state regulation, reproduction, and a wide variety of specific metabolic pathways [3]. The pleiotropic aspects of this essential nutrient have raised the interest of worldwide geneticists and nutritionists in the last few decades. In general, Se functions through inserting the amino acid selenocysteine (Sec) into a multitude of proteins, which are known as selenoproteins. To date, 25 human selenoprotein genes and 24 confirmed avian selenoprotein genes [4] have been identified.

The health impact of upregulation of antioxidant selenoenzymes and cytoprotective properties mediated by Se has been well researched. Dietary Se intakes can affect the synthesis and expression of selenoproteins by modulating status of Se [5]. Appropriate Se status leads to health benefits, such as decreased mortality and enhanced immune function. Moreover, early research demonstrated that both Se deficiency and excess can cause adverse health effects. Impairments in antioxidant protection, redox regulation, immunity response, energy production, infertility, and reproduction are associated with Se deprivation [6]. Se deficiency has been demonstrated to impair growth performance and lead to exudative diathesis [7], which causes economic loss in chicken. Oversupplementation of Se may also have adverse effects on people or animals, commonly called selenosis. It causes decreased laying rate [8] and skin and liver damage in poultry [9].

Se is widely used as a feed additive for enhancing the antioxidants, immunity, and health of chickens, pigs, and other livestock, as well as to obtain Se-enriched byproducts. As the organic form of Se, selenium yeast is commonly used because it is well absorbed and biologically safe [10]. Some studies on broilers and laying hens showed that selenium yeast supplementation aids in maintaining poultry health, maintaining a desirable egg shape index, and supporting shell thickness [8,11], as well as enhancing antioxidant status and organs Se deposition [12]. Previous studies also demonstrated that dietary supplementation of selenium yeast can improve the productive performance in aging hens [13,14]. Selenium as a trace element plays an important role in the production performance and antioxidant capacity in laying hens. Higher absorption and deposition efficiency of selenium yeast may be more conducive to the utilization of selenium in aged laying hens, helping to extend the laying cycle of hens and reduce the breeding costs. We hypothesized that selenium yeast has beneficial effects on egg and shell quality, plasma antioxidants, and selenium deposition in aged laying hens. Moreover, the potential molecular mechanisms of selenium yeast on eggshell strength were explored by using RNA-seq-based global transcriptome analysis.

## 2. Materials and Methods

The experimental animal protocols for this study were approved by the Animal Care and Use Committee of China Agricultural University (No. AW05060202–1).

### 2.1. Animals, Experimental Design, Diets, and Husbandry

A total of 525 76-week-old commercial Jing Hong laying hens were housed in the Poultry Experiment Base of China Agricultural University (Hebei, China) and fed with an average basal Se content of 0.056 ± 0.012 mg/kg corn–soybean diet (the composition and nutrient level of the basal diet are shown in Table 1) for 6 weeks (from 76 to 82 weeks of age) to induce Se depletion in aged laying hens before the selenium yeast (SY) feeding period. After Se depletion, Jing Hong laying hens with similar laying rate (average laying rate = 74.2%) were randomly allocated to 7 treatment groups with 5 replicates (15 chickens per replicate) each. All of the chickens were raised in cages, with three chickens per cage. One group was fed the basal diet only (SD), three groups were supplemented with 0.15, 0.30, or 0.45 mg/kg selenium yeast (SY) (Alltech, Nicholasville, KY, USA), and the other three groups were supplemented with 0.15, 0.30, or 0.45 mg/kg sodium selenite (SS) for 12 weeks (from 83 to 95 weeks of age). The supplemental and analyzed Se levels of each diet are shown in Table 2. The different dose of Se was premixed into the diets, and the corn–soybean diet was formulated to satisfy the Chinese Feeding Standard of Chicken (NY/T33-2004). Batches of the experimental diets were produced every 4 weeks to prevent the feed from mildewing. The hens were housed in an environmentally controlled room maintained at 25 °C and had a daily lighting schedule of 16 h light (from 5 am to 9 pm), 8 h dark (from 9 pm to 5 am).

### 2.2. Data and Sample Collection

Blood samples (8 mL per laying hen) were taken from the main wing vein and collected into an ethylene diaminetetraacetic acid (EDTA) anticoagulant tube every two weeks (at 76, 78, 80, 82, 85, 87, 89, 91, 93, and 95 weeks of age) during the whole 18-week experiment. Plasma was separated by centrifugation at 4 °C, 3000 rpm, for 10 min and stored at −30 °C for further analysis. After Se supplementation for 12 weeks (experimental weeks 18), 2 random laying hens per replicate group were slaughtered 15–20 h post-ovulation. Ovary, magnum, isthmus, and shell gland were sampled and frozen in liquid nitrogen immediately. All samples were stored at −80 °C prior to analysis. In addition, shell gland organs were fixed in 4% paraformaldehyde for hematoxylin–eosin staining.

### 2.3. Measurement of Egg Quality and Translucent Egg

Egg quality was measured during the Se depletion and Se feeding periods. In the Se depletion periods, egg quality was measured at the start and end of this period. In the Se feeding period, egg quality was measured at 6-week intervals; an average of 6 eggs from each replicate in one day were selected randomly for eggshell strength, egg weight, egg yolk color, Haugh unit, and albumen height measurement using a digital egg tester (NABEL, DET-6000) immediately. An average of 6 eggs from each replicate in one day were used for the measurement of the translucent egg at 6-week intervals. The eggs were stored for 0, 7, and 14 days at 20 °C and classified into 4 score levels: 1 means the best shell translucence degree, 4 means the worst shell translucence degree, and the scoring method was referenced in [15].

### 2.4. Determination of Se Content

To determine the Se content of the feed, plasma, ovary, magnum, isthmus, and shell gland, all samples (0.5–1 g for feed, ovary, magnum, isthmus, and shell gland and 0.5–1 mL for plasma) were digested in a mixture of concentrated nitric acid and perchloric (2 HNO_3_:1 HClO_4_) for about 2 h at 200 °C until white fumes appeared. Then, 5 mL hydrochloric acid (1 HCl: 4 H_2_O) solution was added in the mixture and heated until white fumes appeared. After the mixture cooled, 20 mL EDTA was added in the digested sample and pH was adjusted to 1.5–2.0. After this, 3 mL 2,3 DiAminoNaphthalene (DAN) was added in the mixture and heated in the boiled water for 5 min. After the mixture cooled at room temperature, 4 mL cyclohexane was added and the mixture was shaken for 8 min. The supernatant was measured by fluorescence method using a Hitachi 850 fluorescence spectrophotometer (Tokyo, Japan) [16].

### 2.5. Measurement of Antioxidant Enzyme Activity

The activities of glutathione peroxidase (GSH-Px) and total superoxide dismutase (T-SOD) in plasma samples were determined using the detection kits of GSH-Px and T-SOD (Nanjing JianCheng Bioengineering Institute, Nanjing, China). The activity of total anti-oxidation capacity (T-AOC) in plasma samples was measured by T-AOC Kit (KeyGEN BioTECH, Nanjing, China) following the manufacturer’s protocol.

### 2.6. Hematoxylin–Eosin Staining

For histological assessment, three-micrometer sections were stained with hematoxylin and eosin (H&E) to assess whether prolonged selenium supplementation in aged laying hens can lead to selenosis or pathological status in shell gland. The entire images were captured by APERIO CS2 (Leica, Germany).

### 2.7. RNA Extraction, Library Preparation, Sequencing, and RNA-Seq Data Analyses

Total RNA was isolated from the shell gland of 4 laying hens using the RNA Isolated Kit (RN4402, Aidlab Biotechnologies, Beijing, China) according to the manufacturer’s protocol. Quality and quantity measurements of the extracted RNA were performed using a nanophotometer (IMPLEN, Westlake Village, CA, USA) and a qubit fluorometer (Life Technologies, Carlsbad, CA, USA), respectively. RNA integrity numbers (RINs) were determined using the 2100 RNA Nano 6000 Assay Kit (Agilent Technologies, Santa Clara, CA, USA). Paired-end transcriptome sequencing was performed using an Illumina Novaseq6000 sequencing platform at Annoroad Biotechnology Co. Inc. (Beijing, China).

### 2.8. Transcriptome Bioinformatic and Statistical Analysis

Raw transcriptome reads were quality controlled by filtering out low-quality and adaptor sequences by fastp Version 0.20.0 to provide clean data for downstream analysis. Clean reads were mapped using Hisat2 to the reference genome downloaded from Ensembl (https://asia.ensembl.org/index.html, accessed on 5 September 2020). Quantification of gene expression level was conducted by FPKM (fragments per kilobase of exon per million parts mapping) using the featureCounts package, and differentially expressed genes (DEGs) were accessed using DESeq2 R package. The DEGs were identified with the following parameters: *p*-values < 0.05 and the fold change value |log2Ratio| ≥ 1. Gene Ontology (GO) and Kyoto Encyclopedia of Genes and Genomes (KEGG) enrichment analysis were further analyzed in the website DAVID (https://david.ncifcrf.gov/, accessed on 13 October 2020). Short Time-series Expression Miner (STEM) was performed (default parameters) to investigate the dynamic changes in gene expression in shell gland related to different selenium yeast supplementation levels. Weighted gene coexpression network analysis (WGCNA) was performed to identify candidate genes affecting eggshell quality regulated by selenium yeast. Pearson correlation (*p* < 0.05 and |r| > 0.4) was constructed for the relationship between modules and Se content. Common genes in significant modules and DEGs were performed with GO enrichment analysis. Sankey charts exhibited the common pathways between pathways and DEGs to investigate the molecular processes affecting eggshell quality by selenium yeast. The genes in common pathways were investigated for protein–protein interaction (PPI) analysis on the STRING (https://string-db.org/, accessed on 14 November 2020) website (confidence = 0.7).

### 2.9. Quantitative Real-Time PCR (qRT-PCR) Analysis

Expression of mRNA from the transcriptomic analysis was verified by qRT-PCR with cDNA from shell gland samples. Primer Premier (v. 5.0) with default parameters was used to design primers for selected genes (Appendix A). β-actin gene served as a housekeeping gene. qRT-PCR analysis was conducted with a reaction volume of 20 μL containing 10 μL Premix (FP209-2, Tiangen, Beijing, China), 0.6 μL forward and reverse primers, 1 μL cDNA, and 7.8 μL DNase/RNase-free deionized water. The reaction conditions followed the protocols and instructions. The 2-ΔΔCT method was used to quantify the relative changes in gene expression versus those of β-actin from qRT-PCR experiments.

### 2.10. Statistical Analysis

Data are presented as mean ± standard error of the mean (SEM), and data were analyzed with one-way ANOVA, followed by Duncan test (SPSS for Windows, version 25; IBM). Statistical differences were considered significant at *p* < 0.05. The linear or quadratic effect of different level of sodium selenite and selenium yeast effects on egg quality were analyzed using SPSS (SPSS for Windows, version 25; IBM), and orthogonal polynomial contrasts were used to examine whether the responses to different Se levels were linear or quadratic. Phenotypic data presentation and linear correlation analysis was carried out using GraphPad Prism (version 7.0, GraphPad Software Inc, San Diego, CA, USA). Every experiment was repeated at least three times.

## 3. Results

### 3.1. Plasma Se Status and Antioxidant Enzyme Activity

In our study, plasma Se status and antioxidant enzyme activities were measured in aged laying hens during the whole experimental period (Table 3 and Table 4). During the depletion period (experimental weeks 0–6), the plasma Se content significantly declined from 0.237 μg/mL to 0.068 μg/mL (experimental weeks 0–4) (*p* < 0.01) and then remained at a stable low status (experimental weeks 4–6) (Table 3). Moreover, the redox state in plasma was altered by dietary Se deficiency (Table 4). There were concomitant decreases in plasma T-AOC activity, T-SOD activity, and GSH-Px activity in the first two weeks of Se depletion (experimental weeks 0–2) (*p* < 0.01, *p* < 0.01, and *p* < 0.05, respectively), then from experimental weeks 2–6, plasma antioxidant status remained relatively stable. In the supplementation period (experimental weeks 6–18), Se content increased drastically in the first two weeks (experimental weeks 6–8) (*p* < 0.01), with a peak after approximately 6 weeks of supplementation (experimental week 12) for SY supplemented. From experimental weeks 12–18, plasma Se content plateaued in all supplementation groups (Table 3). Se content had a better deposition effect in SY supplementation than SS supplementation. Interestingly, the plasma Se content in the SY0.45 group recovered to the level before depletion after 8-week supplementation (experimental weeks 14), but other treatments did not. Similarly, the plasma antioxidant capacity rose when the Se supplementation diets were fed. There were no significant differences in T-AOC activity after 4 weeks of Se supplementation (experimental weeks 10), but it was higher in the SS and SY groups than the SD group after 8 weeks of supplementation (*p* < 0.01). The T-SOD and GSH-Px activity in plasma after 12 weeks of supplementation (experimental week 18) were significantly higher (*p* < 0.05) in the SY0.45 group than the SD group (Table 4). Compared with the SS and SD groups, SY supplementation groups had a better antioxidant activity in plasma.

### 3.2. Egg Quality and Organs Se Levels

The changes in the egg quality during the selenium deficiency over time are shown in Table 5. With the increasing week of age and Se depletion, egg weight has no significant changes; however, eggshell strength was significantly lower (*p* = 0.04). Albumen height, Haugh unit, and yolk color were also significantly lower (*p <* 0.01). During the supplementation period, eggshell strength was significantly higher in the SY0.45 group than in the SD group at 6 weeks and 12 weeks post supplementation start and higher than the SY0.15 group after 12 weeks of SY supplementation (*p* < 0.05) (Table 6). Eggs collected after 6 and 12 weeks of supplementation (experimental weeks 12 and 18) were classified into four grades according to the degree of shell translucence (Figure 1a and Appendix A). After 6 weeks of SY supplementation, shell translucence score was generally lower in the supplemented groups at all three lengths of egg storage (Figure 1a). In addition, eggs from Se supplemented hens had lower egg weight, albumen height, and Haugh unit after 12 weeks of supplementation (Table 6).

Se status of organs associated with laying performance including shell gland (Figure 1b), eggshell (Figure 1c), magnum (Figure 1d), isthmus (Figure 1e), and ovary (Figure 1f) which were measured after 12 weeks of selenium yeast supplementation (experimental week 18). The results showed that dietary selenium yeast levels correlated in a dose-dependent manner with organs Se levels. Shell gland H&E staining showed no degeneration or necrocytosis of the mucosal epithelium cell layer, and shell gland edema in the uterus for all treatment (Figure 1g).

### 3.3. Gene Expression in the Shell Gland

A total of 1308 DEGs were assessed in the SY groups compared with the SD group (fold change > 1.00, *p* < 0.05) (Appendix A). The upregulated and downregulated DEGs are displayed in Figure 2a. In comparison with the SD hens, 442 DEGs were shared among the three SY supplemented groups, and 296, 144, and 130 specific DEGs were found for the SY0.15, SY0.30, and SY0.45 groups, respectively (Figure 2b). The biological functions of these DEGs were classified by GO term and KEGG pathway enrichment analysis (Figure 2c,d and Appendix A). The GO analysis identified DEGs enriched in 44 biological progress, 8 molecular functions, and 18 cellular compartments relevant to the current work. These include cell cycle processes (regulation of cell proliferation, regulation of epithelial cell proliferation, and extrinsic apoptotic signaling pathway), female organ development (response to estrogen), and developmental processes affected by calcium ions (osteoblast development, positive regulation of bone mineralization, and voltage-gated calcium channel activity). Based on the KEGG pathway enrichment analysis, ECM–receptor interaction, MAPK signaling pathway, TGF-beta signaling pathway, calcium signaling pathway, glycosaminoglycan biosynthesis, and glutathione metabolism were obtained.

The dose effects of Se on shell gland gene expression were performed by STEM. Total DEGs were clustered into 20 profiles, of which three trend profiles (1, 2, and 17) showed significant enrichment (*p* < 0.05) with the colored block (Figure 2e), and 17 profiles without color represented the nonsignificant trends (data not shown). Profile 2 showed that the expression of 155 genes displayed a reducing trend during the low Se level phase. The expression of 136 genes displayed an opposite (rising) trend during the low Se level phase in profile 17. In profile 1, the expression of 46 genes showed an initial decrease but a subsequent increase with the SY0.15 treatment.

### 3.4. Shell Gland Transcriptome Analysis

Egg quality traits are important for laying hens, especially in the late laying period. In our study, eggshell strength in the SY0.45 group was significantly higher than that in the SD and SY0.15 groups. Thus, RNA-seq and bioinformatics were used to explore the potential key genes associated with eggshell strength after selenium yeast supplementation. The Venn diagram showed DEGs from comparisons of SD vs. SY 0.45 and SY0.15 vs. SY0.45 (Figure 3a); 78 DEGs were shared in both two comparisons, including cell migration inducing hyaluronan binding protein (*CEMIP*), syndecan 3 (*SDC3*), solute carrier family 6 member 17 (*SLC6A17*), ovalbumin (*OVAL*), period circadian clock 2 (*PER2*), and proenkephalin (*PENK*). A total of 73 DEGs were unique in the SY0.15 vs. SY0.45 including solute carrier family 13 member 5 (*SLC13A5*), and proopiomelanocortin (*POMC*), and 697 DEGs were unique in the SD vs. SY0.45, including epiregulin (*EREG*), otopetrin 2 (*OTOP2*), Wnt family member 11 (*WNT11*), pleiotrophin (*PTN*), and carbonic anhydrase 2 (*CA2*). To validate the transcriptomic analysis results, the relative gene expression of the candidate gene (*OVAL*, *CEMIP*, *SLC6A17*) was performed by quantitative real-time PCR (qRT-PCR). The qRT-PCR results showed a similar expression pattern compared with RNA-seq results, indicating that these genes were validated (Figure 3b). Functional enrichment analysis results showed that all DEGs were enriched in cell cycle processes (regulation of cell proliferation, p53 signaling pathway, cell cycle, and apoptosis), follicle development (progesterone-mediated oocyte maturation and response to estrogen), eggshell mineralization (calcium signaling pathway, glycosaminoglycan biosynthesis, negative regulation of BMP signaling pathway, and cellular zinc ion homeostasis), and glutathione metabolism (Figure 3c,d). Trend analysis was performed to reveal the expression patterns of candidate DEGs related to eggshell strength at different selenium yeast supplementation levels. Total DEGs were clustered into eight profiles, of which two were significant trend profiles (*p* < 0.05), including profiles 1 and 6 (colored block) (Figure 3e). The expression of 158 genes including *CA2*, *SLC13A5*, and *OTOP2* decreased with higher supplementation levels of SY in profile 1. In profile 6, the expression of 123 genes including *PENK*, *WNT7A*, and *EREG* exhibited an obvious increase with the higher SY supplementation levels.

Network analysis is useful to identify genes that are putative hubs of gene coregulation. Here, WGCNA was used to explore hub genes related to eggshell strength affected by selenium yeast intake. Due to that the eggs collected did not correspond to laying hens for transcriptome sequencing, eggshell strength from each group was not used for WGCNA analysis. Thus, linear correlation was used to analyze the relationship between shell gland Se content with eggshell Se content and eggshell strength (Figure 3f), and Se content in shell gland had a positive relationship with egg strength (*p* = 0.0013) and eggshell Se content (*p* < 0.05). The results of WGCNA analysis showed that these genes were segmented into three significant modules (*p* < 0.05), and for each of these modules, the correlation of the eigengene with shell gland Se content was computed (Figure 3g). The expression of gene in the Memagenta and MEgreen module was positively correlated with the shell gland Se content, and in MEpurple it was negatively correlated (Appendix A). Common genes both in significant modules and DEGs were selected to explain the effect of selenium yeast on eggshell strength. Functional enrichment analysis showed that common pathways were implicated in the regulation of cell cycle, cell migration, eggshell mineralization, reproductive organ development, and shell gland development (Figure 3h). The PPI result of genes in common GO pathways showed that candidate genes in previous results included *PTN*, *SDC1*, *WNT11*, and *PENK* (Figure 3i).

## 4. Discussion

Avian eggshell is made of columnar calcite crystals which protect the eggs from physical damage and microbial contamination and provide a calcium source for the developing embryo [17]. Nevertheless, eggshell quality can begin to deteriorate over time with the characteristics of decreasing eggshell strength, increasing ratio of translucent eggs, and increasing numbers of abnormal eggs, which cause substantial economic losses. Numerous studies have identified that Se is beneficial for health, egg, and shell quality of laying hens [18]. However, the effect of selenium yeast on egg quality and antioxidant activity in aged laying hens has not been elucidated.

Se deficiency and supplementation are closely related to Se status and antioxidants capacity. In general, plasma Se content and antioxidant enzyme are generally considered as useful biomarkers of both Se status and dietary intake [19]. Antioxidant enzyme activity including T-AOC and T-SOD serves as an important index for redox states in livestock and poultry. GSH-Px is a Se-dependent enzyme which has abilities for protecting organs from oxidative damage. In our study, Se deficiency for 6 weeks (experimental weeks 0–6) led to a significant decrease of Se status (71.31% of Se content) and antioxidants capacity (49.60% of T-AOC, 58.31% of GSH-Px, and 16.71% of T-SOD) in aged laying hens. Consistent with other studies, dietary Se deficiency led to a significant decrease of organs Se content, GSH-Px, T-AOC, and SOD [20] activities in plasma. Based on the percent decline of these measured variables, Se content in plasma decreased the most, suggesting that it was more sensitive to Se status than the antioxidant enzymes. In the Se supplementation period, both selenium yeast and sodium selenite supplementation also increased the antioxidant capacity and organs Se status. Consistent with other studies [21,22], Se content in plasma and organs increased with a dose-dependent trend after Se supplementation. However, different sources of Se supplementation do not have the same effect on plasma Se deposition in aged laying hens. Overall, selenium yeast is more effective in supplementing plasma indicators in aged laying hens. After 6 weeks of supplementation (experimental week 12), plasma Se content in the SY0.45 group was recovered, but other groups did not, suggesting that higher dose and better-absorbed Se may be required by aged laying hens. Meanwhile, the activities of GSH-Px, T-AOC, and T-SOD in plasma increased in the Se supplementation period. The data are in agreement with the results previously published showing that antioxidant enzyme activities exhibited a dose-dependent increase with increasing Se supplementation [23], but in our study, some deeper understanding of Se supplementation in aged laying hens was obtained; supplementation with 0.30 mg/kg sodium selenite more consistently restored plasma antioxidant enzyme activity affected by depletion, and in the selenium yeast groups, 0.45 mg/kg selenium yeast supplementation achieved better effects. Overall, selenium yeast supplementation was more beneficial in mitigating the loss of antioxidant enzyme activity in aged laying hens, probably due to the better utilization of organic selenium. Interestingly, it was noticed that the Se deposition hierarchy in organs 15–20 h post-ovulation associated with laying were isthmus, magnum, ovary, and shell gland, successively.

In the late stage of the laying period, changes in nutrient intake have a great impact on laying performance. Studies clearly indicated that selenium yeast plays crucial roles in poultry nutrition and production. Eggshell strength is vital in ensuring the integrity and safety of the egg contents, and it tends to deteriorate with the increasing of bird age [24]. In our study, Se deficiency decreased eggshell strength in aged laying hens. It has been reported that Se deficiency is detrimental to bone microarchitecture, possibly through decreasing antioxidant capacity [25], and bone microarchitecture plays an essential role in eggshell formation. Thus, these results suggest that oxidative stress caused by Se deficiency may decrease eggshell strength in conjunction with age. Moreover, 0.45 mg/kg selenium yeast supplementation for 6 and 12 weeks increased eggshell strength and decreased egg weight and shell translucency; however, the eggshell strength was not found to be improved in sodium selenite supplementation groups in aged laying hens. Consistent with previous studies, eggshell breaking strength was significantly increased after high selenium yeast supplementation [26]. Translucent eggshells are a problematic issue that affects eggshell appearance and decreases the commercial value of eggs. The cause of eggshell translucency is still unknown but is inferred to be associated with variations of the eggshell membrane [27]. The observed decrease in translucent eggshell suggests that Se supplementation for 6 weeks may ameliorate the decline of egg quality. However, after 12 weeks of both sodium selenite and selenium yeast supplementation, there was no significant improvement in translucent eggshell, and these results also suggest that translucent eggshell may not be influenced by different selenium sources. Increasing egg weight in the last phase of a laying cycle is another problematic issue [28] because large eggs are more difficult to handle and are more prone to breaking during transport and collection. The results in our study showed that with increasing weeks of supplementation, high dose of sodium selenite and selenium yeast supplementation decreased the egg weight, suggesting that high dose of sodium selenite and selenium yeast supplementation in the late laying period may play a beneficial role in egg quality. The yeast in selenium-enriched yeast was mainly to provide the organic source of selenium, which was not in an activated state. It did not produce ingredients such as Xylanase that improve bone strength and eggshell quality [29]. The Se intake range which is beneficial for humans and animals is fairly narrow and has been described as a U-shaped curve. A low feeding level was chosen in this experiment, but whether this dose will cause the shell gland of aged laying hens to be in a toxic state still needs to be measured by H&E staining. The results indicated that 0.45 mg/kg selenium yeast supplementation for 12 weeks (experimental week 18) did not cause pathological lesion of aged laying hens.

In our study, the effect of selenium yeast supplementation on shell gland of aged laying hens was obtained by whole transcriptome analysis. The results of transcriptome analysis revealed several novel genes and biological pathways regulating ion transport, eggshell calcification, and, consequently, the eggshell formation. Based on the enrichment pathways, molecular functions, and gene expression trends, thirteen genes were identified as potential candidate genes during eggshell formation, including *CEMIP*, *SDC3*, *OVAL*, secreted phosphoprotein 1 (*SPP1*), *SLC6A17*, *SLC13A5*, *OTOP2*, *CA2*, *POMC*, *PTN*, *PENK*, *WNT11*, and *EREG*.

Calcium carbonate crystals interact with the shell organic matrix to form a highly ordered microstructure during shell calcification. Genes involved in shell mineralization have attracted our attention, including *CEMIP*, *SDC3*, *OVAL*, and *SPP1*. *CEMIP* is involved in glycosaminoglycan metabolism and calcium release from the endoplasmic reticulum [30]. The upregulated *CEMIP* expression (log_2_ fold change > 2 for SD vs. SY0.45 and SY0.15 vs. SY0.45) showed that selenium yeast supplementation promoted gene expression for calcium release needed for eggshell formation. SDC3 protein possesses domains containing some potential glycosaminoglycan attachment sites [31]. Glycosaminoglycan is widely thought to regulate mineral deposition and determine the properties of eggshell [32]. Thus, it is inferred that the upregulation of *SDC3* may play an active role in eggshell formation. *OVAL* is an abundant eggshell matrix protein binding calcium, and it plays an active role in carbonate formation [33]. In coherence with our results, the upregulation of *OVAL* might improve eggshell formation. *SPP1*, also known as osteopontin, has mineral-binding domains [34] which are involved in calcium metabolism and calcium carbonate precipitation [32] for eggshell calcification. Consistent with other studies, the upregulation of *SPP1* occurred during eggshell calcification [35]. Similarly, DEGs and enriched pathways, including calcium signaling pathway, glycosaminoglycan biosynthesis and positive regulation of bone mineralization, were also closely associated with eggshell calcification and formation. Hence, the results suggest that selenium yeast supplementation has a role in eggshell calcification by regulating the expression of *CEMIP*, *SDC3*, *OVAL*, *SPP1*, and other candidate genes beneficial for eggshell formation.

The eggshell formation process requires a large amount of calcium (Ca^2+^) and bicarbonate (HCO_3_^−^); thus, ion transport plays a crucial role. Some DEGs found in molecular pathways associated with ion transport were screened in our study, including voltage-gated calcium channels and calcium signaling pathways. *SLC6A17* and *SLC13A5* genes are part of the solute carrier family and play an important role in transporting ions across cell membranes for synthesis of eggshell. It has been reported that *SLC6A17* may regulate alanine transport during eggshell formation, while *SLC13A5* plays an important role in the initiation of eggshell synthesis [36]. The carbonate ions required for eggshell synthesis are produced by carbonic anhydrase (CA) reactions, and the uterine glandular cells possess the carbonic anhydrase activity sites. There is growing evidence that high expression levels of *CA2* expression play a pivotal role in the conversion of intracellular CO_2_ to HCO_3_^−^ in chickens [37]. As a member of the otopetrin gene family, the *OTOP2* gene may have similar functions to *OTOP1*, which is regarded as a modulator of cellular calcium influx [38], to transport calcium across the uterine epithelium for eggshell calcification [39]. Our findings inferred that *SLC6A17*, *SLC13A5*, *OTOP2*, and *CA2* are regulators of ion transport in the shell gland and are affected by selenium yeast supplementation.

The eggshell formation also depends upon numerous physiological adaptations and processes by the uterine cells, as well as reproductive hormones. In our study, functional enrichment analysis showed that selenium yeast supplementation may affect the response to estrogen, regulation of cell proliferation, regulation of epithelial cell proliferation, and extrinsic apoptotic signaling pathway. *POMC* plays a role in stimulating the release of cortisol hormone and its upregulated expression during the mineralization period [39]. Consistent with other studies, a higher expression of *POMC* may result in a hard eggshell [37]. *PTN* is a developmentally-regulated growth factor and its expression is induced by estrogen [40]; *PTN* and estrogen were reported to have a pivotal role in eggshell formation [36]. These findings suggest that selenium yeast supplementation may affect the expression of reproductive hormones and, subsequently, eggshell formation. Indeed, WGCNA analysis also identified some candidate genes which affected the eggshell formation regulated by selenium yeast supplementation, including *PENK*, *WNT11*, and *EREG*. These upregulation DEGs were reported in previous studies to have a vital role in eggshell formation [36,37].

Calcium (Ca) is one of the key nutrients for bone formation and eggshell quality [41], but all ingredients except Se were consistent in our treatment; thus, the content of Ca was the same for each treatment. The effect of different Se levels on the utilization and deposition of calcium deserves further discussion. Previous studies also found that trace elements manganese and zinc enhanced eggshell strength by improving biosynthesis of glycosaminoglycan [42] and affecting carbonic anhydrase activity [43], respectively. An increasing number of studies have explored the fact that selenium yeast has a positive effect on eggshell quality of laying hens [18]. However, studies on the molecular mechanism of selenium yeast affecting eggshell quality are limited. To date, it is hypothesized that selenium exhibits beneficial effects on eggshell quality and may be directly involved in the process of regulating eggshell formation or the interaction between trace elements. Overall, based on biological functions of the DEGs, it is hypothesized that the molecular pathways impacted by selenium supplementation on aged laying hens are those related to eggshell mineralization, specifically regulation of shell calcification and ion transduction. Despite that, additional studies are needed to understand the molecular mechanisms modulated by selenium yeast supplementation in the various phases of eggshell formation.

## 5. Conclusions

Dietary selenium yeast supplementation at a dose of 0.45 mg/kg improved eggshell strength and reduced eggshell translucence in aged laying hens. Moreover, observed increases in plasma Se content, plasma antioxidant enzyme activity, and reproductive organs Se content were dependent on dietary selenium yeast supplementation level. Transcriptomic analysis revealed that selenium may play a role in eggshell quality through regulating the expression of genes involved in eggshell mineralization and ion transport. *CEMIP*, *OVAL*, *SLC6A17*, *SLC13A5*, *POMC*, and *PENK* were identified as candidate genes affecting eggshell formation via regulation by selenium.

## Figures and Tables

**Figure 1 animals-13-00902-f001:**
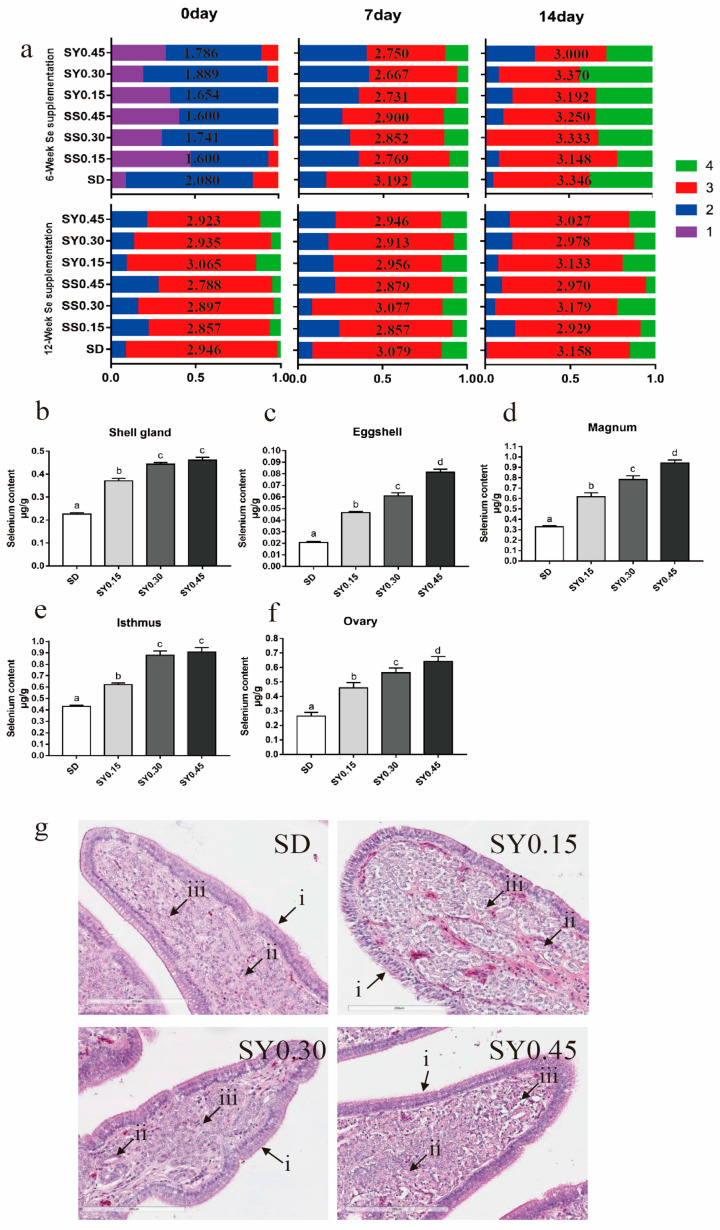
Eggshell quality and organs Se status in aged laying hens during the whole experiment period. (**a**) The changes of translucent eggs during selenium yeast supplementation, the average score is shown in the middle of the bar. The data in the horizontal row were detected on different days when eggs were stored; similarly, the data in the vertical row were detected in different weeks of age. (**b**) Selenium content in the shell gland after selenium yeast supplementation for 12 weeks. (**c**) Selenium content in eggshell after selenium yeast supplementation for 12 weeks. (**d**) Selenium content in magnum after selenium yeast supplementation for 12 weeks. (**e**) Selenium content in isthmus after selenium yeast supplementation for 12 weeks. (**f**) Selenium content in ovary after selenium yeast supplementation for 12 weeks. Data are expressed as mean ± SEM. Within each panel, means without a common letter differ at *p* < 0.05. (**g**) H&E staining of representative in shell gland after selenium yeast supplementation for 12 weeks (20×), (i) the mucosal epithelial cell layer, (ii) shell gland, (iii) lymphocyte. Scale bar = 200 μm.

**Figure 2 animals-13-00902-f002:**
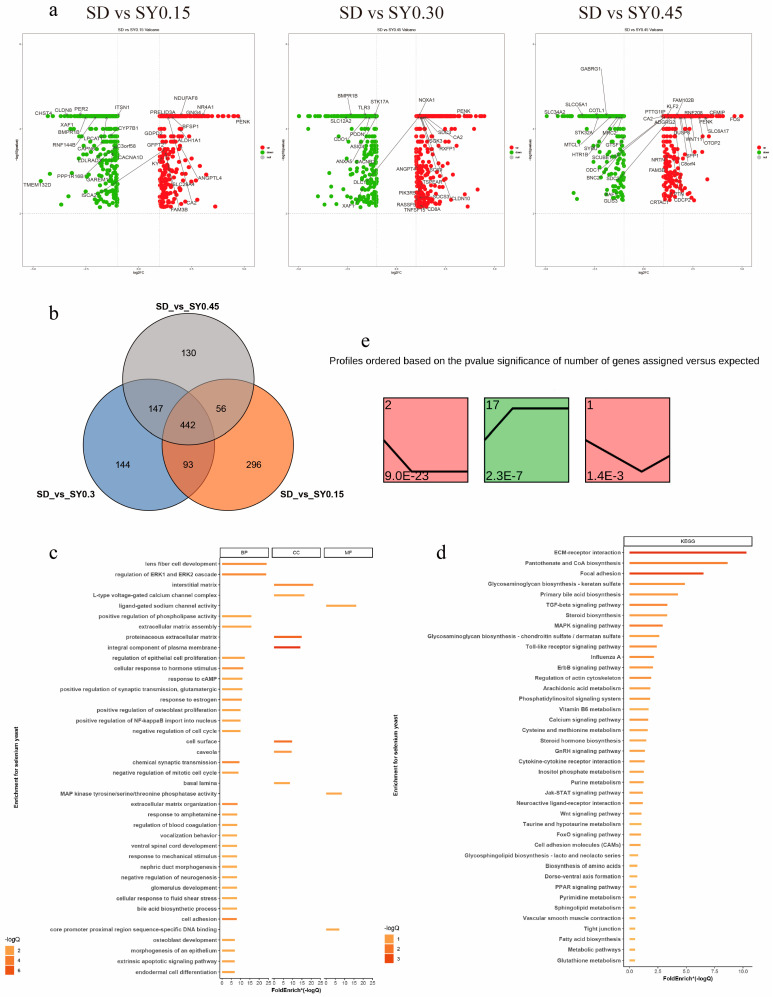
Transcriptome analysis revealed the effect on gene expression of shell gland in aged laying hens. (**a**) Scatter plots of DEGs (SD vs. SY0.15, SD vs. SY0.30, and SD vs. SY0.45). Red points represent upregulated genes with a log2 (fold change) > 1 and *p* < 0.05. Blue points represent downregulated genes with a log2 (fold change) < −1 and *p* < 0.05. Gray points represent genes showing no significant difference. Fold change = normalized gene expression in the SY0.15, SY0.30, and SY0.45 group/normalized gene expression in the SD group. (**b**) Venn diagram showing the distribution of DEGs in three comparisons of Se supplementation. The number of DEGs is indicated in the diagram. (**c**) Gene Oncology enrichment analysis of DEGs identified among the three conditions (SD vs. SY0.15, SD vs. SY0.30, and SD vs. SY0.45). (**d**) KEGG pathway enrichment analysis of DEGs identified among the three conditions (SD vs. SY0.15, SD vs. SY0.30, and SD vs. SY0.45). (**e**) Identification of 3 significant gene cluster profiles by STEM. Colored block trend: significant enrichment trend (*p* < 0.05). The number of genes in each significant cluster is shown after the cluster number.

**Figure 3 animals-13-00902-f003:**
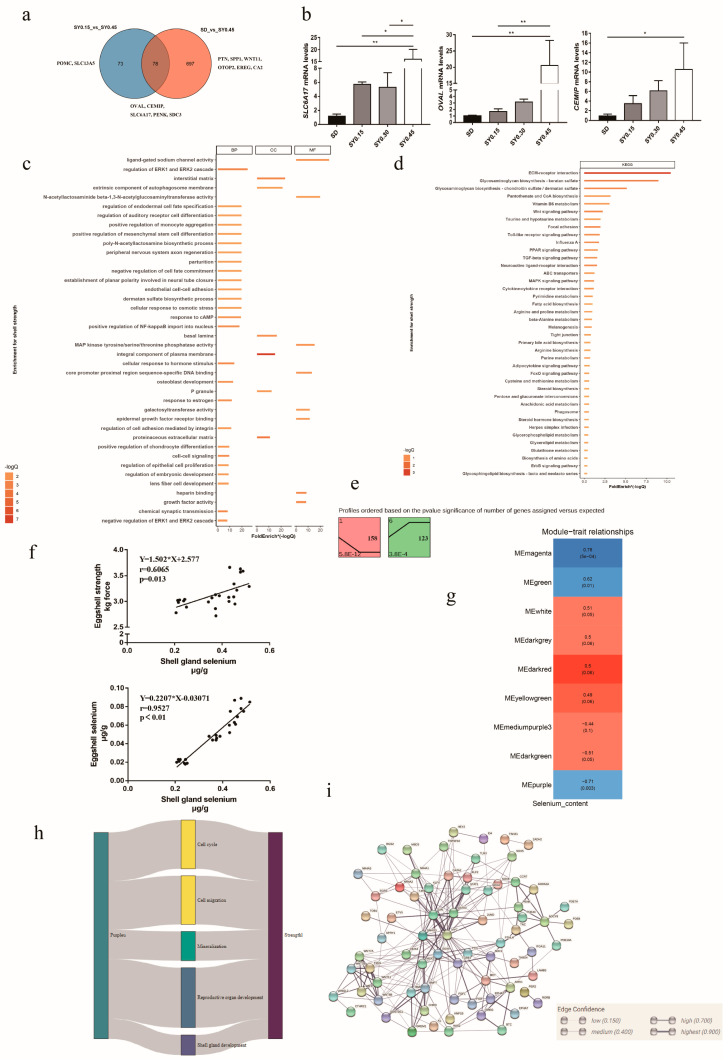
The potential molecular mechanism of selenium yeast affecting the eggshell strength in aged laying hens. (**a**) Venn diagram showing the distribution of DEGs associated with eggshell strength. (**b**) Validation of DEGs associated with eggshell strength using qRT-PCR, * means *p* < 0.05 and ** means *p* < 0.01. (**c**) Gene Oncology enrichment analysis of DEGs associated with eggshell strength (SY0.15 vs. SY0.45 and SD vs. SY0.45). (**d**) KEGG pathway enrichment analysis of DEGs associated with eggshell strength (SY0.15 vs. SY0.45 and SD vs. SY0.45). (**e**) Trend analysis of DEGs associated with eggshell strength expression. Colored block trend: significant enrichment trend (*p* < 0.05). (**f**) The linear correlation between shell gland selenium content with eggshell strength and eggshell selenium content. (**g**) WGCNA analysis identified specific transcriptional modules related to eggshell strength in aged laying hens after selenium yeast supplementation. (**h**) Sankey charts were performed among differential expressed genes in modules and pathways. (**i**) The PPI network of common genes in modules and DEGs.

**Table 1 animals-13-00902-t001:** Composition and nutrient of the basal diet.

Items	Content (%)	Nutrient Level	Content (%)
Corn	60.50	AME, Mcal/kg	2.57
Soybean meal	21.60	Crude protein (%)	15.0
Wheat	4.10	Lysine (%)	0.74
Cottonseed meal	2.00	Methionine (%)	0.3
Soybean oil	0.50	Methionine + Cystine (%)	0.55
Calcium carbonate (GR)	9.50	Calcium (%)	3.7
Calcium phosphate (GR)	1.00	Available phosphorus (%)	0.31
*DL*-methionine	0.08	Total phosphorus (%)	0.54
Phytases	0.015	Selenium ^c^ (mg/kg)	0.056
Vitamin premix ^a^	0.035		
Mineral premix ^b^	0.15		
Sodium chloride (GR)	0.30		
50% Choline chloride	0.10		
Experimental additives	0.12		
Total	100.00		

Abbreviations: GR = guaranteed reagent; ^a^ provided per kilogram of diet: vitamin A, 12,500 IU; vitamin D3, 2500 IU; vitamin E, 18.75 mg; vitamin K3, 2.65 mg; vitamin B1, 2 mg; vitamin B2, 6 mg; vitamin B12, 0.025 mg; biotin, 0.325 mg; folic acid, 1.25 mg; niacin, 50 mg; ^b^ provided per kilogram of diet: Cu, 8 mg; Fe, 80 mg; Zn, 80 mg; Mn, 60 mg; I, 1.2 mg. Selenium in each treatment group is shown in the experiment design in Table 2; ^c^ selenium in the basal diet is measured value.

**Table 2 animals-13-00902-t002:** Dietary selenium levels.

	Control	Selenium Yeast			Sodium Selenite		
Group	SD	SY0.15	SY0.30	SY0.45	SS0.15	SS0.30	SS0.45
Measured values ^1^	0.056	0.211	0.377	0.522	0.205	0.267	0.480
SEM	0.012	0.015	0.049	0.030	0.016	0.049	0.030

Abbreviations: SD = selenium deficiency; SY0.15 = 0.15 mg/kg selenium yeast group; SY0.30 = 0.30 mg/kg selenium yeast group; SY0.45 = 0.45 mg/kg selenium yeast group; SS0.15 = 0.15 mg/kg sodium selenite group; SS0.30 = 0.30 mg/kg sodium selenite group; SS0.45 = 0.45 mg/kg sodium selenite group; ^1^ the values were measured by fluorescence method, and the unit is mg/kg.

**Table 3 animals-13-00902-t003:** Selenium content of plasma during the whole experiment period (experimental weeks 0–18).

Items	SD	SS0.15	SS0.30	SS0.45	SY0.15	SY0.30	SY0.45	SEM	*p*-Value
Se content 0 week	0.24							0.01	
Se content 2 week	0.10							0.01	
Se content 4 week	0.07							0.01	
Se content 6 week	0.05							0.01	
Se content 8 week	0.05 ^a^	0.12 ^b^	0.18 ^d^	0.19 ^d^	0.15 ^c^	0.20 ^d^	0.23 ^e^	0.01	<0.01
Se content 10 week	0.04 ^a^	0.15 ^b^	0.18 ^bc^	0.17 ^bc^	0.15 ^bc^	0.19 ^cd^	0.24 ^d^	0.01	<0.01
Se content 12 week	0.05 ^a^	0.16 ^b^	0.19 ^bc^	0.19 ^bc^	0.16 ^b^	0.22 ^cd^	0.24 ^d^	0.01	<0.01
Se content 14 week	0.07 ^a^	0.17 ^b^	0.18 ^b^	0.19 ^bc^	0.18 ^b^	0.22 ^c^	0.26 ^d^	0.01	<0.01
Se content 16 week	0.09 ^a^	0.18 ^b^	0.18 ^b^	0.22 ^c^	0.20 ^bc^	0.22 ^c^	0.26 ^d^	0.01	<0.01
Se content 18 week	0.08 ^a^	0.16 ^b^	0.19 ^b^	0.20 ^bc^	0.19 ^b^	0.23 ^cd^	0.26 ^d^	0.01	<0.01

Abbreviations: SD = selenium deficiency; SY0.15 = 0.15 mg/kg selenium yeast group; SY0.30 = 0.30 mg/kg selenium yeast group; SY0.45 = 0.45 mg/kg selenium yeast group; SS0.15 = 0.15 mg/kg sodium selenite group; SS0.30 = 0.30 mg/kg sodium selenite group; SS0.45 = 0.45 mg/kg sodium selenite group; ^a–e^ values within a row with different superscripts differ significantly at *p* < 0.05, and the unit of the selenium content is μg/mL.

**Table 4 animals-13-00902-t004:** Antioxidants capacity of plasma during whole experiment period (experimental weeks 0–18).

Items	SD	SS0.15	SS0.30	SS0.45	SY0.15	SY0.30	SY0.45	SEM	*p*-Value
GPx 0 week	3550.15							96.41	
GPx 2 week	2082.02							155.19	
GPx 4 week	1605.42							253.39	
GPx 6 week	1480.00							208.94	
GPx 10 week	2102.40	2323.20	3316.00	2904.00	2720.00	2893.33	3184.00	139.45	0.31
GPx 14 week	2111.60 ^a^	2874.58 ^b^	3430.51 ^c^	3486.15 ^c^	3480.65 ^c^	3458.10 ^c^	3502.11 ^c^	105.73	<0.01
GPx 18 week	2340.10 ^a^	2791.19 ^ab^	3421.85 ^b^	3151.45 ^ab^	3210.45 ^ab^	3377.27 ^b^	3545.17 ^b^	123.47	0.10
T-SOD 0 week	270.61							10.44	
T-SOD 2 week	236.91							8.79	
T-SOD 4 week	250.81							6.39	
T-SOD 6 week	225.37							11.42	
T-SOD 10 week	248.73	240.42	253.51	230.07	269.56	247.52	253.91	6.51	0.83
T-SOD 14 week	251.36	252.82	257.31	257.60	267.29	244.79	255.16	6.22	0.99
T-SOD 18 week	257.87	261.67	269.36	294.32	270.94	275.39	299.94	5.17	0.62
T-AOC 0 week	3.72							0.49	
T-AOC 2 week	2.64							0.37	
T-AOC 4 week	1.88							0.16	
T-AOC 6 week	1.87							0.13	
T-AOC 10 week	1.91	2.72	3.15	2.67	2.52	2.92	3.13	0.20	0.23
T-AOC 14 week	1.48 ^a^	3.17 ^b^	3.84 ^b^	2.93 ^b^	3.18 ^b^	3.50 ^b^	5.02 ^c^	0.21	<0.01
T-AOC 18 week	2.09 ^a^	3.44 ^ab^	4.04 ^bc^	3.94 ^bc^	4.41 ^bc^	5.34 ^c^	5.12 ^c^	0.27	<0.01

Abbreviations: SD = selenium deficiency; SY0.15 = 0.15 mg/kg selenium yeast group; SY0.30 = 0.30 mg/kg selenium yeast group; SY0.45 = 0.45 mg/kg selenium yeast group; SS0.15 = 0.15 mg/kg sodium selenite group; SS0.30 = 0.30 mg/kg sodium selenite group; SS0.45 = 0.45 mg/kg sodium selenite group; ^a–c^ values within a row with different superscripts differ significantly at *p* < 0.05, and the unit of the selenium content is U/mL.

**Table 5 animals-13-00902-t005:** Egg quality of aged laying hens fed selenium-deficient diets over time.

Items	Egg Weight (g)	Eggshell Strength (kg Force)	Albumen Height (mm)	Haugh Unit (U)	Yolk Color
Before depletion	61.78	3.26	6.53	79.10	6.89
After depletion	63.83	2.89	5.63	71.46	5.90
SEM	1.07	0.17	0.22	2.01	0.99
*p*-value	0.06	0.04	<0.01	<0.01	<0.01

**Table 6 animals-13-00902-t006:** Egg quality of aged laying hens after selenium supplementation.

Items	SD	SS0.15	SS0.30	SS0.45	SS-Linear *p*-Value	SS-Quadratic *p*-Value	SY0.15	SY0.30	SY0.45	SY-Linear *p*-Value	SY-Quadratic *p*-Value	SEM	Treatment *p*-Value
Experimental weeks 12													
Egg weight (g)	65.30	64.09	64.02	62.56	0.05	0.90	63.59	63.94	62.27	0.39	0.44	0.36	0.38
Eggshell strength (kg force)	2.70 ^a^	2.80 ^ab^	3.16 ^ab^	2.99 ^ab^	0.08	0.38	2.96 ^ab^	2.95 ^ab^	3.28 ^b^	0.08	0.26	0.06	0.13
Albumen height (mm)	6.25	5.89	6.05	5.54	0.26	0.88	6.27	5.97	5.54	0.10	0.39	0.09	0.47
Haugh unit (U)	76.09	73.01	74.36	70.64	0.15	0.01	76.51	73.69	70.74	0.08	0.98	0.86	0.43
Yolk color	5.80	5.87	5.76	6.04	0.27	0.37	5.82	5.97	6.12	0.09	0.98	0.05	0.31
Experimental weeks 18													
Egg weight (g)	67.82 ^a^	65.59 ^abc^	66.25 ^ab^	63.76 ^c^	<0.01	0.77	63.76 ^c^	65.40 ^bc^	65.13 ^bc^	0.22	0.31	0.30	<0.01
Eggshell strength (kg force)	2.98 ^a^	3.15 ^ab^	3.16 ^ab^	2.84 ^a^	0.05	0.07	2.88 ^a^	3.08 ^ab^	3.48 ^b^	<0.01	0.57	0.05	0.03
Albumen height (mm)	7.15 ^a^	6.62 ^b^	6.30 ^bc^	5.97 ^c^	<0.01	0.45	6.29 ^bc^	6.70 ^b^	6.28 ^bc^	0.97	0.06	0.06	<0.01
Haugh unit (U)	81.66 ^a^	79.26 ^ab^	78.31 ^ab^	74.22 ^c^	<0.01	0.42	76.83 ^bc^	78.27 ^ab^	77.31 ^bc^	0.81	0.49	0.47	<0.01
Yolk color	5.90	6.06	5.88	6.15	0.19	0.60	5.96	5.98	6.18	0.15	0.49	0.04	0.26

Abbreviations: SD = selenium deficiency; SY0.15 = 0.15 mg/kg selenium yeast group; SY0.30 = 0.30 mg/kg selenium yeast group; SY0.45 = 0.45 mg/kg selenium yeast group; SS0.15 = 0.15 mg/kg sodium selenite group; SS0.30 = 0.30 mg/kg sodium selenite group; SS0.45 = 0.45 mg/kg sodium selenite group; a–c values within a row with different superscripts differ significantly at *p* < 0.05.

## Data Availability

The datasets generated and analyzed in the current study are available in the Sequence Read Archive (SRA) database at NCBI under BioProject ID PRJNA716513.

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
