# Peer review of "Effects of Selenium Yeast on Egg Quality, Plasma Antioxidants, Selenium Deposition and Eggshell Formation in Aged Laying Hens"

_animals, 2023, doi:10.3390/ani13050902_

Round 1

Reviewer 1 Report

Lines 82 to 84 – when you write “Numerous studies on broilers and laying hens have shown that selenium yeast supplementation aids in maintaining poultry health, maintaining a desirable egg shape index, supporting shell thickness…” and cite just three references (8, 11, 12), it is not numerous. I recommend two corrections:

1.       Cite more papers to subside your phrase or change “numerous” for “some” and It is better to focus on laying hens

Line 98 - Jing Hong laying hens are a regional genetic line ? brown or white eggs ? what is the longevity of them ? in other commercial lines the hens can produce until 100 weeks of age.

You told us that the laying rate was 74.2%, and I ask if you consider this rate as regular for this age ?

Line 116 - Table 1

the amount of AME in MJ/kg seems very low or typed incorrectly. Please, review this number

Please, inform in Table 1 which nutrient level was calculated and which one is analysed, please. I undersood that just selenium was determined, and the other ones were calculated

Total Aminoacids or Digestible Aminoacids ?

Methionine+Cysteine (%) or Methionine+Cystine (%) ?

0,31 is related to Available phosphorus (%) or Non-phytate phosphorus (%) ? Did you consider as a synonym ? If yes, choose just one name, please. Total phosphorus and Non-phytate phosphorus can be determined in a laboratory.

Lines 128 to 130

“Blood samples (8 mL per laying hen) were taken from the main wing vein and collected into an Ethylene diaminetetraacetic acid (EDTA) anticoagulant tube every two weeks during the whole 18-week experiment” – please, specify which ages and if you collected before the experiment.

Lines 137 to 149

Measurement of Egg Quality and Translucent Egg:

-          Internal variables - egg weight, egg yolk color, Haugh unit, and albumen height

-          External variables - Eggshell Thickness and eggshell translucence degree

Eggshell Thickness by Eggshell Thickness Gauge (which equipment (Trade mark) ? in mm ?

Eggshell translucence degree – is it a visual score as proposed by Wang et al (2019) ?

In my opinion, you should increment it. You must correlate it with common data as eggshell percentage, specific density and calcium content of the eggshells. Why did not you measure it ? I read your introduction and I did not see any clear correlation with selenium nutritional effects on the specific variables you measured.

Line 215

Correct “seleniu yeast” for “selenium yeast”

Results

Lines 223 to 225

In the phrase “It is well recognized that plasma and serum selenium reflect the Se status of body and that decreased expression of systemic selenoproteins caused by dietary Se deficiency leads to oxidative stress and oxidative stress-related diseases [17].” you cite a reference. In my opinion, this is not correct and in this item you must describe the results, without discussion at this moment. You can delete this phrase or move to discussion.

In the text, I was looking for the sodium selenite results, and I did not find it in all tables and figures in the main document, especially in Figure 1 (line 250), Figure 2 (line 283). In table 2, you analyzed the selenium contents of seven diets and just show results from selenium yeast ? You confirmed it in the item “2.1. Animals, Experimental Design, Diets, and Husbandry, in which you detailed in the lines 105 to 109 that “One group was fed the basal diet only (SD), three groups were supplemented with 0.15, 0.30, or 0.45 mg/kg selenium yeast (SY) (Alltech, Nicholasville, KY, USA), and other three groups were supplemented with 0.15, 0.30, or 0.45 mg/kg sodium selenite (SS) for 12 weeks (from 83 to 95 weeks of age)”.

I have to ask why all the results were not completely presented ? I did not understand why you selected some results to show in a complete form. Is it a mistake or I did misunderstand it ? Because of this, I did not understand if you analyses are complete or not. This is crucial for your paper complete comprehension. I do not feel comfortable to read a paer in which you present part in a main document and other ones in supplements.

Lines 243 to 246

You describe that “There were no significant differences in T-AOC activity after 4 weeks of Se supplementation (experimental weeks 10), but it was higher in the SS and SY groups than the SD group after 8-weeks of supplementation (P < 0.01) (Figure 1b and Supplement Table 3)”, but these data are not available for both in the main paper, why ? In my opinion, it is important to show all data together to represent your original design.

Lines 248 and 249

The same happens as described above. SS group data are not available here

Lines 257 and 258

The changes in the egg quality during the selenium deficient over time are shown in Supplement Table 4. I really do not understand why you show data in a supplement and not in the main paper. Sorry, but, if the information is important, it should be published together.

Lines 258 and 259

Tendencies are not feasible. Decide if it is significant or not. It you determined that 5% is the statistical probability, you should say if it is different or no. (“With the increasing week of age and Se depletion, egg weight 258 trended to be higher (P = 0.06).”)

Line 283

Figure 2 must be reorganized and show separately. There is too much information for just one figure. It is not didactic to explain your results. In addition, the sodium selenite data are not presented in this figure. Some data are presented in the Figure 2 and part of it in Table 3. Why you did not merge same quality data in the same table or figure ?

Lines 322 and 379 – Figures 3 and 4

It was very difficult to observe the results of Figure 3. I tried to zoom in on the figure on the computer, but it is impossible to express any opinion regarding the results with minimal viewing. I understand that the volume of information is quite large, but unfortunately the figure needs to have visibility and clarity. Without that, it was not possible for me to analyze the results.

Conclusion

You said that 0,45 mg/kg selenium yeast is enough to improve eggshell quality, but I believe that 0,15 mg/kg had good results too. Please, be clearer in this conclusion. Selenium yeast is good for eggshell, and we recommend the supplementation of 0,45 mg/kg

In general, the paper is very consistent, but, in my opinion, it focuses more on selenium yeast and forget sodium selenite as a supplement. The authors must be attempted to attend the main purpose of the experiment, that was to evaluate selenium supplementation and the source of selenium. The other point is that we need to open two or three documents to evaluate the results because the authors divided it. Of course, I understand that it is important to make all the data available to the researchers, but this make more difficult to evaluate the general results. The authors can use these resources, as the editors allows it, but, in my opinion, the sodium selenite results must be showed in the main paper and not just in the additional files. This paper must be re-written to allows that the original design of the experiment was respected. It is not adequate to discuss 90% of the paper with one source when you tested two.

The figures must be reorganized, because they are full of information and can be divided to become more visible. The transcriptome and metabolome information are so difficult to read because the letters are very difficult because of the unreadable letters.

Reviewer 2 Report

This article is generally interesting and well written. My minor comments aim to increase the scientific soundness and clarity of it. Additionally, some language ambiguities must be corrected.

Line 39 - From histological point of view there are only four kinds of tissues: epithelial, connective, muscular and nervous. Therefore, such terms as “reproductive tissue” (line 39, line 521) “shell gland tissue” (line 135), “tissues associated with laying performance” (line 271) are not justified. The authors terribly confuse organs with tissues. Organs are assembled from the four basic types of tissues and have cells with specialized functions. Additionally, the term “tissue Se status” is awkward. In fact, it is Se concentration in different organs.

Line 41 – please explain what Se stands for (as used for the first time).

Line 74 – “physiological diseases” is kind of oxymoron.

Line 88-89 – Please explain your hypothesis properly.

Line 168 – this is repetition of sentence in lines 135-136. What parameters were assessed/measured during HE staining? In other words what was purpose of it? Please explain what changes are seen in Figure 2h?

Line 260 – how “yolk color” can be lower?

Figure 3a, 3c, 3d are too small to read.

Line 418-421 – The authors did not discuss that beside microelements/micronutrients like Se or Ca, also other components (diet for example) can influence the eggshell as well as bones quality. It is believed that organic Se is derived from any feed ingredient including wheat, maize (with or without the addition of hybrid rye) and soybean meal. Also supplements like xylanse substantially improves bone strength and eggshell quality (see recent work of MuszyÅ„ski et al. Assessing Bone Health Status and Eggshell Quality of Laying Hens at the End of a Production Cycle in Response to Inclusion of a Hybrid Rye to a Wheat–Corn Diet. Vet. Sci. 2022, 9, 683. https://doi.org/10.3390/vetsci9120683). This issue is definitely in line with the topic of the current article and in Reviewers opinion the authors should briefly address this issue in the Discussion as well as acknowledge this work.

Round 2

Reviewer 1 Report

The authors did some corrections in the body of the paper, but I really do not understand why you decided to show some results just for selenium yeast and supressed some selenium selenite results. It is not feasible for my understanding. I know that you have the total information in the additional files, but I see more focus on the yeast in the paper. My suggestions were not completely attended in this final version.

Reviewer 2 Report

The authors reasonably answered my concerns.

Author Response

We are thankful to the reviewer for nice suggestions and comments.